# Learning Road Scene-level Representations via Semantic Region Prediction

**Zihao Xiao, Alan Yuille**
Department of Computer Science
Johns Hopkins University
`{zxiao10,ayuille1}@jhu.edu`

**Yi-Ting Chen**
Department of Computer Science
National Yang Ming Chiao Tung University
`ychen@cs.nycu.edu.tw`

**Abstract:** In this work, we tackle two vital tasks in automated driving systems, i.e., driver intent prediction and risk object identification from egocentric images. Mainly, we investigate the question: what would be good road scene-level representations for these two tasks? We contend that a scene-level representation must capture higher-level semantic and geometric representations of traffic scenes around ego-vehicle while performing actions to their destinations. To this end, we introduce the representation of semantic regions, which are areas where ego-vehicles visit while taking an afforded action (e.g., left-turn at 4-way intersections). We propose to learn scene-level representations via a novel semantic region prediction task and an automatic semantic region labeling algorithm. Extensive evaluations are conducted on the HDD and nuScenes datasets, and the learned representations lead to state-of-the-art performance for driver intention prediction and risk object identification.

**Keywords:** Semantic Region Prediction, Egocentric Vision, Driver Intent, Risk Object Identification

## 1 Introduction

For automated driving systems (e.g., advanced driver assist systems, ADAS) to navigate highly interactive scenarios, they must be able to perceive states of traffic elements, forecast traffic situations, identify potential hazards, and plan the corresponding actions. The field has made substantial progress in the past few years [1, 2, 3, 4, 5, 6, 7, 8, 9, 10, 11, 12, 13, 14, 15, 16, 17, 18, 19, 20]. In this work, we focus on improving the performance of driver intention prediction [21, 22, 23] and risk object identification [24, 25, 26, 27, 28] from egocentric videos. Solving both tasks from egocentric videos is crucial for safety systems such as ADAS, where front-facing cameras are the primary device.

Existing works for both tasks [22, 24, 26, 27, 28] utilize image annotations of the tasks (intent prediction and potential hazard identification) and object cues from object detection to train networks in a supervised learning manner. Additionally, the authors of [22, 29] leverage temporal models such as LSTM [30] and ConvLSTM [31], and spatial-temporal interaction between traffic participants are modeled using spatial-temporal graph [23] and graph convolutional networks [32] to further improve the performance of tasks. While promising results are demonstrated, the learned representations are ineffective to the trained task. Moreover, the representations only encode road scenes in the nearby locality. We contend that a scene-level representation must capture higher-level semantic and geometric representations of traffic scenes around ego-vehicles while performing actions to their destinations in order to reason about the larger scenes. We introduce a novel representation called the semantic region, as shown in Fig. 1. Semantic regions are areas where ego-vehicles visit while taking an afforded action to their destination. The birth of semantic region is motivated by road affordance (i.e., possible actions that a vehicle can take in an environment). For instance, while turning left at an intersection, the vehicle visits the semantic regions (in yellow), i.e., the *crosswalk* near the ego vehicle, *the area of intersection*, and the *crosswalk* on the left sequentially. If different afforded actions are taken, different semantic regions will be visited. Note that different road topologies (e.g., 3-way intersections and straight roads) afford different actions. Our insight

6th Conference on Robot Learning (CoRL 2022), Auckland, New Zealand.

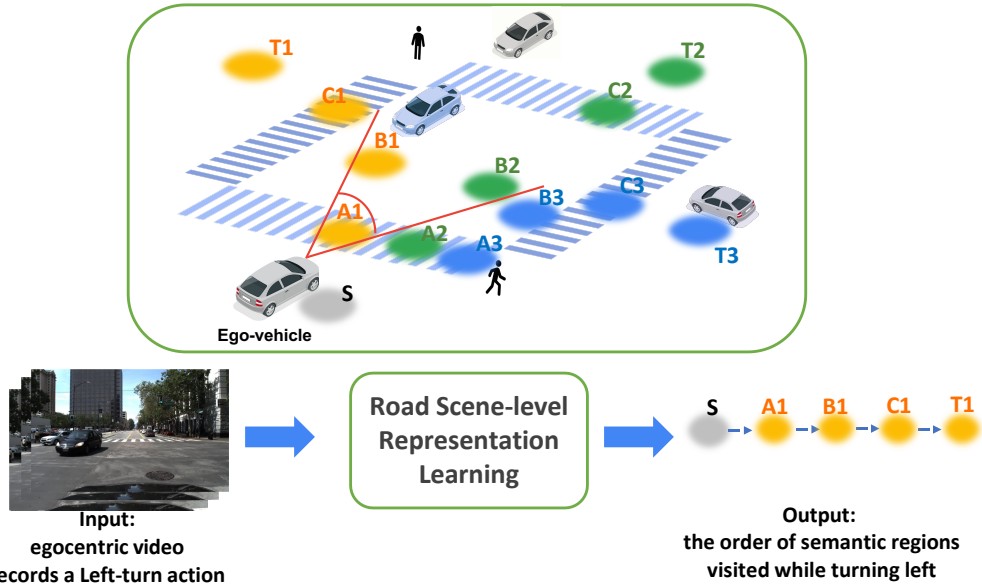

Figure 1: **Main idea.** We contend that a scene-level representation must capture higher-level semantic and geometric representations of traffic scenes around ego-vehicles while performing actions to their destinations in order to reason about the larger scenes. We propose semantic region, a novel representation that represents areas where ego vehicles visit while taking an afforded action. We associate egocentric images, representing views from different locations of road scenes under specific actions, with the corresponding semantic regions. We cast road scene-level representation learning as semantic region prediction and demonstrate the learned representations are effective for driver intention prediction and risk object identification.

is that there are finite regions that a vehicle visits when taking action afforded the underlying road topology. Therefore, we associate egocentric images, representing views from different locations of road scenes under certain actions, to the corresponding semantic regions.

We cast scene-level representation learning as semantic region prediction. Specifically, the model predicts future semantic regions sequentially, given historical observations before turning left. For instance, as shown in Fig. 2a, given egocentric images representing semantic regions $S$ and $A_1$, the task aims to predict future semantic regions $B_1$, $C_1$, and $T_1$ in sequential order. To enable representation learning, we design an automatic semantic region annotation strategy to label every egocentric image collected in intersections with the corresponding semantic region, which reduces the annotation burden.

We demonstrate the effectiveness of the scene-level representation learning framework on driver intention prediction [22] and risk object identification [27]. We achieve superior performance compared to strong baselines for driver intention prediction on the HDD dataset [33]. Furthermore, we show favorable generalization capability without additional training on nuScenes [34]. Moreover, our framework obtains state-of-the-art performance for risk object identification. Specifically, we boost the current best-performing algorithm [27] by 6%.

Our contributions are summarized as follows. First, we propose a novel representation called semantic region, which aims to capture higher-level semantic and geometric representations of traffic scenes around ego vehicles while performing actions to their destination. Second, we cast scene-level representation learning as semantic region prediction (SRP) and propose an automatic labeling algorithm for intersections to reduce annotation burdens. Third, we conduct extensive evaluations on the HDD and nuScenes datasets to prove that the effectiveness of the learned representations leads to significant improvements in driver intention prediction and risk object identification.

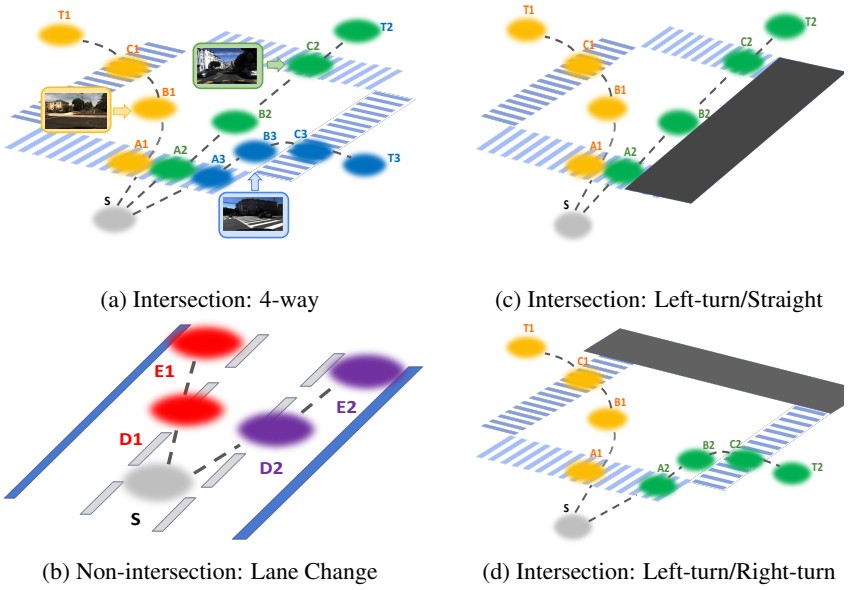

(a) Intersection: 4-way

(c) Intersection: Left-turn/Straight

(b) Non-intersection: Lane Change

(d) Intersection: Left-turn/Right-turn

Figure 2: **Semantic regions in different types of road topology.** Semantic regions are areas where ego-vehicles visit while taking actions afforded the underlying road topology. For instance, at 4-way intersections, three actions (i.e., Left-turn, Straight, Right-turn) are afforded. Given egocentric images while performing an afforded action, we associate them with the corresponding semantic regions. In this work, we cover a wide range of road topologies, i.e., 4-way/3-way intersections and straight roads with multiple lanes.

## 2 Related Work

**Driver Intention Prediction.** Advanced driver-assistance systems predict driver intention [21, 22, 35, 36, 37, 38] to avoid potential hazards. Doshi et al., [21] predict driver's intent via reasoning distances to lane markings and vehicle dynamics for driver intention prediction in highway scenarios. In the Brain4Car project [22, 39, 23], multi-sensory signals, including GPS and street maps, are used for anticipation of driving maneuvers. Similarly, pre-computed road topology maps around intersections are utilized to extract features such as ego position and dynamics, distance to surrounding traffic participants, and legal actions at the upcoming intersection to predict driver intention [35]. Recently, Casas et al., [37] leverage rasterized HD maps as input deep neural networks for intent prediction. Instead of formulating intent prediction as a recognition problem, Hu et al., [38] formulate intention prediction as entering an *insertion area* defined on a pre-computed road topology map. For instance, if the intent is turning left, the corresponding *insertion area* is $T_1$ as shown in Fig. 2a. Unlike existing methods exploiting pre-computed road topology, we learn road scene-level representations via semantic region prediction, which captures higher-level semantic and geometric representations of traffic scenes around ego vehicles while performing actions to their destinations. We empirically demonstrate the value of learned representations.

**Risk Object Identification.** The goal of risk object identification is to identify object(s) that impact ego-vehicle navigation [24, 40, 41, 25, 26, 42, 43, 27]. The authors of [24, 25, 43] construct datasets with object importance annotation, and supervised learning-based algorithms are designed and trained to identify risk/important objects. The task can be formulated as selecting regions/objects with high activations in visual attention heat maps learned from end-to-end driving models [40, 26, 42]. Recently, Li et al., [27] formulate risk object identification as a cause-effect problem [44]. They propose a two-stage risk object identification framework and demonstrate favorable performance over [40, 26]. In this work, we extend [27] with the learned road scene-level representations because driver intention is crucial for risk/important object identification [25]. Note that they [25] assume that the planned path is given. In this work, we tackle a more challenging setting where driver intention is unknown and should be inferred from egocentric images.

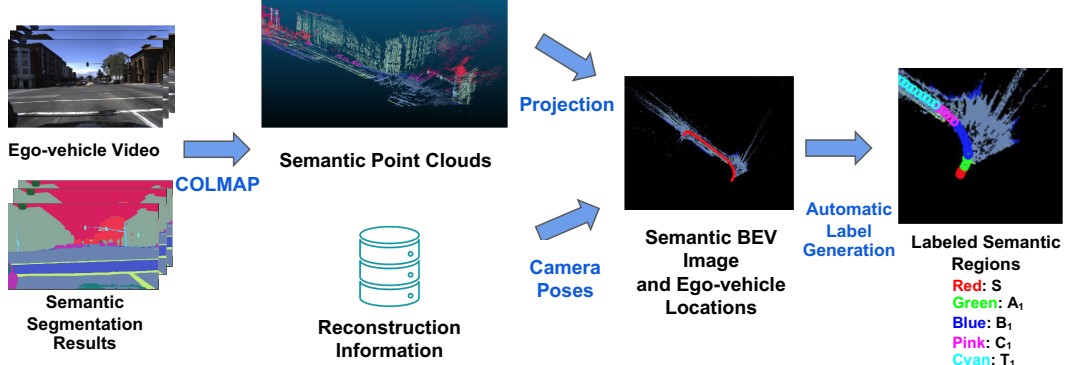

Figure 3: **Automatic labeling of semantic regions in intersections.** We show an example of generating labels of semantic regions from a *Left-turn* egocentric video sequence. The semantic regions are consistent with the ones in Fig. 2a. Best viewed in color.

# 3 Automatic Semantic Region Labeling

We propose an automatic labeling strategy to ease the burdens. The overall generation process is depicted in Fig. 3. Specifically, a three-step strategy is proposed. First, given egocentric videos collected while taking afforded actions (i.e., *Left-turn*, *Straight*, *Right-turn*, and *Lane-change*) without interacting with traffic participants from the HDD dataset [33][1], we apply COLMAP [45], to obtain a dense 3D reconstruction and camera poses. In addition, semantic segmentation [46] is applied to every egocentric image. Second, each 3D point is projected onto images so that the point is visible to obtain the corresponding semantic candidates. Then, a simple winner-take-all strategy is used to determine the final label. We project the semantic 3D point cloud to the ground plane to obtain a semantic Bird-Eye-View (BEV) image. Third, we label the semantic region of each camera pose with the information from semantic BEV image. For example, in intersections, we assume that ego vehicles will visit two *crosswalks* sequentially while taking afforded actions. Camera poses that overlap with the first *crosswalk* and the second *crosswalk* are denoted as $A_i$ and $C_i$, respectively. The poses located between $A_i$ and $C_i$ are $B_i$. Camera poses located before the first *crosswalk* and the second *crosswalk* as $S_i$ and $T_i$, respectively. Each index $i$ represents an afforded action. Last but not least, while the results of COLMAP and semantic segmentation are generally well, we use two additional criteria to select good samples: 1) 3D reconstruction is successful, and 2) reconstructed camera poses form a coherent trajectory. Note that the algorithm is in general applicable for different topologies. However, we observed failures for *lane-change* in non-intersection due to inaccurate 3D reconstruction. Therefore, we manually annotate videos that ego-vehicles perfrom lane-change. Details of automatic semantic region labeling are provided in the supplementary materials.

# 4 Methodology

In this section, we discuss the details of road scene-level representation learning from egocentric video via semantic region prediction. In addition, we illustrate how to transfer the learned representation to two downstream tasks, i.e., driver intention prediction and risk object identification.

## 4.1 Scene-level Representation Learning via Semantic Region Prediction

We contend that a scene-level representation must capture higher-level semantic and geometric representations of traffic scenes around ego-vehicle while performing actions to their destinations. Thus, we proposed the representation called semantic region, which is a high-level abstraction of road affordance. We expect a model capturing the association between the temporal evolution of egocentric views and semantic regions. To this end, We cast the egocentric road scene affordance representation learning as a semantic region prediction task. We build our Semantic Region Prediction (SRP) cell based on TRN cells [47]. The STA (spatio-temporal accumulator) in the TRN cell

---

[1]The HDD dataset provides large-scale annotations of afforded actions.

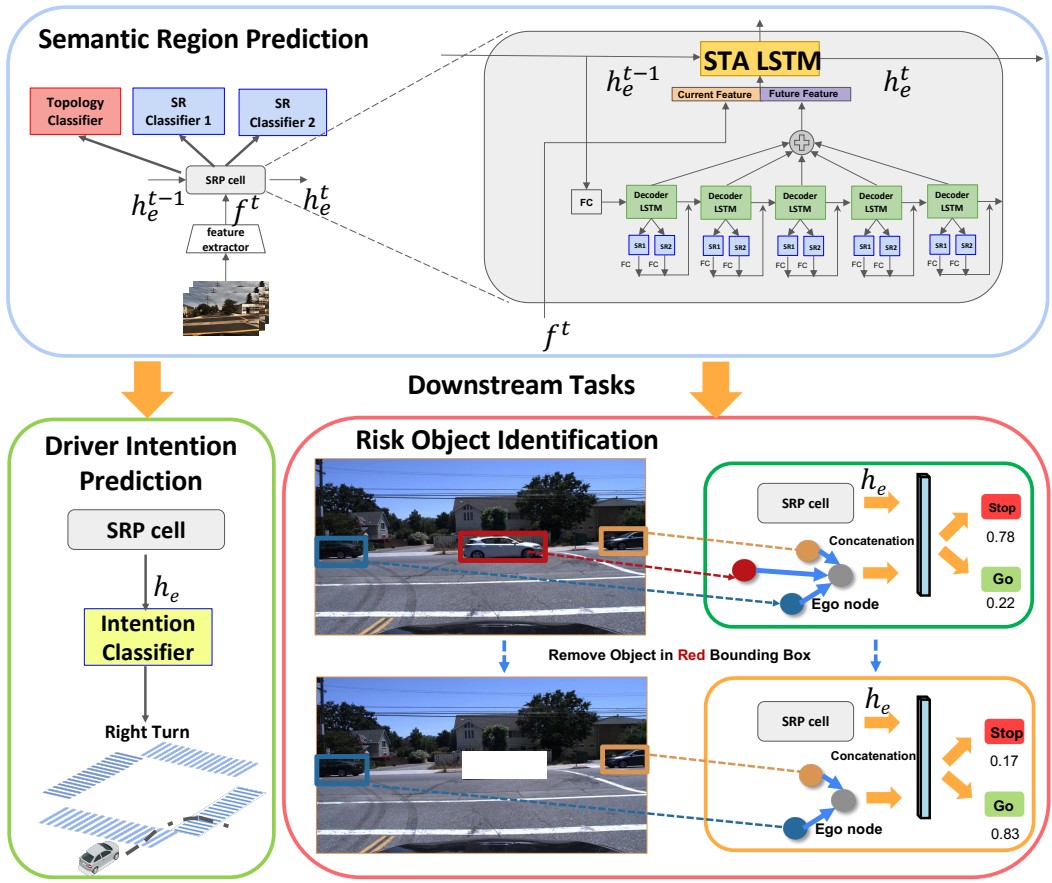

Figure 4: **The proposed network architecture for semantic region prediction and models for downstream tasks.** We propose to learn road scene representation via Semantic Region Prediction (SRP). The hidden state of the SRP cell serves as road scene representation and is utilized in two downstream tasks: driver intention prediction and risk object identification.

makes use of predicted future cues from the temporal decoder and the accumulated historical information to form better action representations. To have a unified framework, We make the following changes to TRN cells. First, we replace the action classifier with two semantic region classifiers for both intersections and non-intersections. Second, in the decoder, the predicted logits of two semantic region classifiers are fused into the input of the next time frame after increasing dimensions with fully connected(FC) layers. Third, we add a topology classifier to determine whether the ego-vehicle is at intersections (4-way, 3-way) or non-intersections (straight road or curve). Note that our design is similar to CILRS [48], a command-conditional imitation learning framework. In our case, we select the corresponding set of semantic regions (as shown in Fig. 2) based on the prediction of topology type.

With SRP cells, our network takes $t_e$ historical frames as input. For each frame, topology type (i.e., whether it is in an intersection), the current semantic region as well as $t_d$ future semantic regions are predicted. We have separate semantic region classifiers for intersections and non-intersections. During training, we only compute losses for the one that matches the ground truth topology type. The semantic region prediction loss $\mathcal{L}$ is defined as

$$\mathcal{L} = \sum_t^{t_e} \mathbf{l}(z_t, o_t) + \sum_{i=0}^{1} \mathbb{1}_{o_t=i} \Big( \sum_t^{t_e} \mathbf{l}(y_t^{i,e}, s_t^i) + \frac{1}{t_d} \sum_{m=0}^{t_d} \mathbf{l}(y_t^{i,d}, s_{t+m}^i) \Big) \tag{1}$$

where $i \in \{0,1\}$ denotes the $i$th semantic region classifier, $z_t$ denotes the topology prediction, and $y_t^{i,e}$ and $y_t^{i,d}$ are the semantic region prediction based on the hidden state of STA and decoder respectively for topology classifier $i$. $\mathbf{l}$ is the Cross-entropy loss, $\mathbb{1}$ is the indicator function, $o_t$ is the

ground truth topology type, and $s_t^i$ is the ground truth of semantic regions derived from Section 3. The overall architecture is depicted in Fig. 4. In practice, it is not necessary to observe all semantic regions in one video clip. For instance, in an intersection, when there are no crosswalks, $A_i$ and $C_i$ will not be presented. For a left-turn vehicle at intersections without a crosswalk, the semantic region sequence will be $S - B_1 - T_1$. The hidden state $\mathbf{h}_e^t$ of STA contains rich information about the road scene. Next, we will show how to incorporate the learned representation into downstream tasks.

## 4.2 SRP-guided Driver Intention Prediction

We follow the definition of anticipation in [39] to define driver intention prediction. Formally, given an sequence of egocentric observations $\{\mathbf{x}^1, \mathbf{x}^2, ..., \mathbf{x}^t\}$, our goal is to predict the future intention $\mathbf{y}_{int}^T$, where $T > t$. Driver intention prediction benefits downstream applications like risk assessment [49]. There are 5 different types of intentions in our setting (i.e., Left-turn, Straight, Right-turn, Left-lane-change, Right-lane-change). We add an intention classifier on top of the hidden state of the STA, $\mathbf{h}_e^t$ in SRP,

$$\mathbf{y}_{int}^T = \mathrm{softmax}(\mathbf{W}_{int}^\top \mathbf{h}_e^t + \mathbf{b}_{int}) \tag{2}$$

where $\mathbf{W}_{int}^\top$ and $\mathbf{b}_{int}$ are the weight and bias terms in the intention classifier, respectively. We name the driver intention prediction model SRP-INT.

## 4.3 SRP-guided Risk Object Identification

The risk object identification task was first introduced in [27]. A Risk object is defined as the one influencing the behavior of the ego-vehicle most in each frame. Given an egocentric video $\{\mathbf{x}^1, \mathbf{x}^2, ..., \mathbf{x}^t\}$, the goal of risk object identification is to output $\{\mathbf{b}^1, \mathbf{b}^2, ..., \mathbf{b}^t\}$, where $\mathbf{b}^j$, $j \in [1, t]$ is the bounding box of the risk object in the $j$-th frame. The authors of [27] proposed a two-stage framework to solve the problem. In the first stage, they trained an object-level manipulable model to predict the driver behavior by incorporating partial CNNs [50]. In the second stage, they iterated through the risk object candidate list and intervened in the input video to simulate scenarios without the presence of a candidate. The simulated scenarios were passed into the driver behavior model. The object causing the maximum driving behavior change was their risk object prediction. The ego-representation in [27] takes a very important role because it captures the information from the image frame and the messages from all the objects. The representation in time $t$, i.e., the last time step, can be written as

$$\mathbf{g}_e^t = \mathbf{g}_f^t \oplus \frac{1}{N}\sum_{k=1}^N \mathbf{g}_k^t \tag{3}$$

where $\mathbf{g}_f^t$ is the representation of the image frame, $\mathbf{g}_k^t, k \in [1, N]$ in the representation for each object, $\oplus$ indicates a concatenation operation, and $\mathbf{g}_e^t$ is the final ego-representation in [27].

We propose SRP-ROI by fusing SRP with the model in [27]. We argue that road scene-level information can benefit the risk object identification task, and propose an SRP-guided representation:

$$\mathbf{g}_e^t = \left((\mathbf{W}_{ego}\mathbf{g}_f^t \oplus \frac{1}{N}\sum_{k=1}^N \mathbf{g}_k^t) + \mathbf{b}_{ego}^t\right) \oplus \boldsymbol{h}_e^t \tag{4}$$

where $\mathbf{W}_{ego}$ and $\mathbf{b}_{ego}$ are the weights and bias terms of a fully connected layer respectively. We follow the two-stage framework in [27] and evaluate our SRP-ROI model on two challenging dynamic risk object categories: crossing vehicles and crossing pedestrians.

# 5 Experiments

## 5.1 Semantic Region Prediction

**Data Collection and Annotation.** We collect video clips of Left-turn, Straight, Right-turn, Left-lane-change, and Right-lane-change from the HDD dataset to train our semantic region predictor. For each video clip, we manually label the topology type. Labels of semantic regions at the intersections are automatically generated with the methods proposed in Section 3. The semantic regions for

| Metric | Intersection | | Non-intersetion | |
|---|---|---|---|---|
| | Current SR | Future SR | Current SR | Future SR |
| Micro Avg Pre | 47.0 | 52.7 | 65.3 | 62.9 |
| Macro Avg Pre | 20.9 | 20.3 | 50.4 | 53.8 |
| mAP | 26.4 | 24.5 | 51.2 | 53.8 |

Table 1: **Performances of Semantic Region Prediction.** Current SR stands for the current semantic region, while Future SR stands for the future semantic region.

| Model | Aux | HDD | | | HDD Interactive | | | nuScenes | | |
|---|---|---|---|---|---|---|---|---|---|---|
| | | Macro Avg Pre | Micro Avg Pre | mAP | Macro Avg Pre | Micro Avg Pre | mAP | Macro Avg Pre | Micro Avg Pre | mAP |
| LSTM[30] | - | 45.0 | 64.9 | 51.5 | 30.8 | 56.2 | 62.4 | 37.3 | **68.8** | 62.0 |
| LSTM-EL[39] | - | 45.0 | 65.5 | 52.4 | 29.1 | 51.8 | 60.9 | 35.6 | 62.0 | 61.0 |
| OadTR[52] | - | 35.9 | 24.3 | 36.3 | 48.4 | 46.9 | 54.1 | **47.8** | 64.3 | 50.7 |
| TRN-Tra | Tra | 45.0 | 70.8 | 47.5 | 30.9 | 59.8 | 57.2 | 35.7 | 58.8 | 58.7 |
| SRP-INT | SR | **55.3** | **73.8** | **57.9** | **67.0** | **70.3** | **69.5** | 41.1 | 68.3 | **66.7** |

Table 2: **Quantitative results of driver intention prediction.** We compare SRP-INT with baselines. **Aux** stands for auxiliary tasks. Tra and SR stand for trajectory and semantic region, respectively. All models have the same feature extractor [46].

non-intersections are annotated by humans. For each video clip, we apply a sliding-window method to obtain training samples. For each sample, we have annotations including topology type, current, and future semantic region labels.

**Implementation Details and Results.** We leverage ResNet50 [23] pre-trained on Mapillary Vistas [45] dataset as the feature extractor. Our SRP takes $l_e$ = 3 historical frames as input. For each frame, $l_d$ = 5 future semantic regions, as well as topology type, are predicted. As shown in Fig. 2a and Fig. 2b, the number of semantic regions in intersection and non-intersections are 13 and 5, respectively. We use Adam optimizer [51] with default parameters, a learning rate of 0.0001, and weight decay of 0.0005. The model is trained for 60 epochs. We train the model with the loss function in Eq. (1). The performances are shown in Table 1. Macro Average Precision, Micro Average Precision, and mAP are chosen as the evaluation metrics.

## 5.2 Driver Intention Prediction

**Testing Data and Experiment Setup.** After training SRP on the video clips in Section 5.1, we further use the intention labels to train the intention classifier. Details are provided in the supplementary materials. We evaluate driver intention prediction models on both HDD [33] test set and nuScenes [34] datasets. Note that in HDD, there is no overlap between the training data and test data. We evaluate models on 1438 sequences in HDD (including 393 interactive scenarios) and 221 sequences in nuScenes. We use the same evaluation metrics as Section 5.1.

**Baselines and Comparisons.** We implement several baselines with the same image feature extractor as the proposed SRP-INT. LSTM [30] is a general-purpose sequential modeling methods. OadTR [52] takes advantage of the popular Transformers [53] and is a competitive online/real-time action recognition model. We also implement LSTM with Exponential Loss (LSTM+EL), as [39] shows the effectiveness of Exponential Loss for driver intention prediction. We modify TRN [47] to predict trajectories (similar to the work [54]) and use the learned representation for intention prediction. As shown in Table 2, we demonstrate favorable performances on both datasets and prove the effectiveness of our framework empirically. Qualitative results are presented in the supplementary materials.

| Model | Crossing Vehicle | | | Crossing Pedestrian | | |
|---|---|---|---|---|---|---|
| | Acc 0.5 | Acc 0.75 | mAcc | Acc 0.5 | Acc 0.75 | mAcc |
| [27] (paper) | 49.2 | 48.6 | 43.0 | 35.7 | 32.1 | 27.0 |
| [27] (our implementation) | 49.2 | 48.2 | 42.7 | 33.3 | 29.8 | 26.2 |
| SRP-ROI | **51.8** | **51.1** | **45.1** | **42.9** | **39.3** | **33.3** |

Table 3: **Quantitative results of risk object identification.** We evaluate risk object identification models on two risk object categories: Crossing Vehicle and Crossing Pedestrian.

### 5.3 Risk Object Identification

**Experimental Setup and Evaluation.** We follow the experiment setup in [27] and train separate models on two challenging dynamic risk object categories: Crossing Vehicle and Crossing Pedestrian. Like [27], we evaluate our models by calculating the IOU between the predicted risk object and ground truth. We report accuracy at IOU thresholds of 0.5, 0.75, and mean accuracy.

**Implementation Details.** We utilize Mask R-CNN [55] and DeepSORT [9] to compute the tracking proposals of risk object candidates. The pre-trained semantic region representation is fused with the ego representation in [27] after passing through a fully connected layer. In practice, the output dimension of the fully connected layer is 100. In this stage, we train the model using Adam [51] optimizer with default parameters, a learning rate of 0.0001, and weight decay of 0.0001. The model is trained for 20 epochs. After training, we follow the inference procedure in [27] to obtain the bounding boxes of the risk object in each frame. We do not apply any heuristic to remove objects from tracking proposals and models are trained separately for each category.

**Quantitative Results.** We compare our method with [27]. The quantitative results show that our model obtain favorable performance compared to [27], which demonstrates that semantic region prediction can help risk object identification. Qualitative results are presented in the supplementary materials.

## 6 Limitations

Although we have shown the effectiveness of our proposed representation, some limitations need further exploration. First, our proposed semantic regions cannot be applied to complicated topologies like roundabouts or other real-world edge cases in intersections. Possible solutions are: defining the semantic regions of all intersections by the number of branches of the intersections and considering one roundabout as a series of 3-way intersections. Second, learning semantic regions from egocentric view images alone is challenging. Additionally, the performance of semantic region prediction at intersections is unsatisfactory. To improve the performance, we could consider incorporating Bird-Eye-View representation [56]. Third, we have not truly associated images with semantic regions. Instead of predicting the label of semantic regions, we could consider an encoder-decoder based model to predict the current/future scene representations [57].

## 7 Conclusion

In this work, we study the problem of road scene-level representation learning from egocentric videos for driver intention prediction and risk object identification. We propose a novel representation called semantic region, which aims to capture higher-level semantic and geometric representations of traffic scenes around ego vehicles while performing actions to their destination. We cast representation learning as semantic region prediction and propose an automatic semantic region labeling algorithm for egocentric videos collected in intersections. We demonstrate the effectiveness of the learned representation on real-world datasets, i.e., HDD and nuScenes. In particular, the learned representation can generalize to unseen data (i.e., nuScenes dataset) without finetuning the driver intention prediction task. We hope that our findings will pave the way for further advances in road scene-level representation learning from egocentric views for downstream tasks such as planning and decision-making.

**Acknowledgments**

A part of the work was done when Z. Xiao was an intern and Y.-T. Chen was a research scientist at Honda Research Institute USA, San Jose, CA, USA. The work is partly sponsored by Honda Research Institute USA. Yi-Ting Chen is supported in part by the Higher Education Sprout Project of the National Yang Ming Chiao Tung University and Ministry of Education (MOE), and the Ministry of Science and Technology (MOST) under grants 110-2222-E-A49-001-MY3 and 110-2634-F-002-051.

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
