# OpenReview forum: "Learning Road Scene-level Representations via Semantic Region Prediction"
_robot-learning.org/CoRL/2022/Conference — CoRL 2022 Poster_

### Official Review · Reviewer_4hg8 · 2022-07-29

**Originality:** Good
**Technical Quality:** Very Good
**Clarity Of Presentation:** Very Good
**Impact:** 3

**Recommendation:**

Weak Accept: I recommend accepting the paper, but will not argue for my recommendation if the majority of other reviewers have a different opinion.

**Summary:**

This paper designs a novel scene-level represent learning framework by predicting the current and future semantic region. The authors utilize the hidden state in the decoding process for two important downstream tasks: driver intent prediction and risk object identification. The experimental results demonstrates the effectiveness of the framework.

**Issues:**

The definition of the semantic region should be described clearer.

**Quality Of The Limitations Section:**

Limitations are addressed clearly

**Reviewer Expertise:**

4: The reviewer is confident but not absolutely certain that the evaluation is correct

**Robotics Focus:**

Highly relevant to robotics but no hardware experiments

**Strengths And Weaknesses:**

Strengths:
1.The paper designs a novel scene-level represent learning framework a by predicting the current and future semantic region.
2.The authors design an automatic algorithm to label the semantic regions.
3.The proposed framework could handle different types of road topology.
4.The representation of the road could improve the precision on downstream tasks.

Weaknesses:
1.The framework separately study the representation of the intersection and non-intersection road topology, which leads to higher cost and parameter quantity in complicated scenario where exists many different types of road topology.
2.The quality of the representation is only verified on two simple downstream tasks, the authors could test it on some driving simulator or real world scenario.

**Summary Of Recommendation:**

This paper designs a novel scene-level represent learning framework by predicting the current and future semantic region. The authors design an automatic algorithm to label the semantic regions and train the represent learning framework. The proposed framework only takes the egocentric frames as input, and could handle different types of road topology. The performance improvement of the two downstream tasks (driver intent prediction and risk object identification) demonstrates the quality of the scene-level representation.  However, there are some minor defects of the paper: The framework separately study the representation of the intersection and non-intersection road topology, which leads to higher cost and parameter quantity in complicated scenario where exists many different types of road topology; The quality of the representation is only verified on two simple downstream tasks, the authors could test it on some driving simulator or real world scenario.

---

> ### Author Response · Authors · 2022-08-21
> **Response to Reviewer 4hg8 (part 1 of 2)**
>
> Thanks a lot for your insightful comments and suggestions! We will address your concerns point-by-point in detail.
>
> **Q1** How can the proposed framework handle higher cost and parameter quantity?
>
> **A1** We agree that cost and parameter quantity will get higher when there exist diverse types of road topology. That is the nature when we go into real-world applications. For example, a parameterized representation of road structures (e.g., [1]) requires 60+ parameters to model intersection-like road structures. To eliminate a parameter space, we look into road affordance, i.e., possible actions that a vehicle can take in an environment. For instance, a 4-way intersection affords left-turn, right-turn, and go-straight. Therefore, we represent an afforded action using a sequence of semantic regions, as shown in Figure 2. Such representations consider both higher-level semantics (including actionable cues) and geometrics of roads.
>
> Recently, predicting semantic birds-eye-view (BEV) from egocentric images [2,3] has attracted great attention. These non-parametric methods have started to show their potential for downstream motion planning [3,4]. However, the performance of semantic BEV is still not satisfactory. Therefore, We hope that our findings will pave the way for further advances in road scene-level representation learning from egocentric views for downstream tasks such as planning and decision making.
>
> [1] Wang et al., A Parametric Top-View Representation of Complex Road Scenes, CVPR 2019
>
> [2] Roddick and Cipolla, Predicting Semantic Map Representations from Images using Pyramid Occupancy Networks, CVPR 2020
>
> [3] Philion and Fidler,  Lift, Splat, Shoot: Encoding Images from Arbitrary Camera Rigs by Implicitly Unprojecting to 3D, ECCV 2020
>
> [4] Chen and Krahenbuhl, Learning from All the Vehicles, CVPR 2022
>
> **Q2** What are the semantic regions in a complicated scenario where exists many different types of road topology?
>
> **A2** In the real-world application, the driving vehicles will encounter different types of road topology. We would like to gently argue that our semantic region predictors can still work properly. Consider a case where an ego-vehicle turns left at a Left-turn/Straight intersection and then turns right at a 4-way intersection, the semantic region will switch from T1 to S right after ego-vehicles exit the Left-turn/Straight intersection. After that, the semantic region will gradually switch from S to T3 when the ego-vehicle makes a right turn. Another difficulty in the real world is that there are more complicated intersections as well as other unusual types of typology. Our current model cannot deal with these situations, as we mentioned in the limitation section. However, we are thinking about the following ways to solve the problem:
>
> 1. defining the semantic regions of all intersections by the number of branches of the intersection.
>
> 2. considereing one roundabout as a series of 3-way intersections.
>
> **Q3** Why does the author only evaluate two simple tasks? The authors could test it on some driving simulator or real-world scenario.
>
> **A3** We would like to highlight that driver intention prediction and risk object identification are two critical and challenging tasks for vision-based intelligent driving systems. For intent prediction, predicting a driver’s intent from egocentric images is challenging. For risk object identification, the setting of [27] is challenging as models have to identify risk objects in an unsupervised manner. While the proposed method outperforms the baselines, the performance is unsatisfactory, as reported in Tables 2 and 3. The authors believe significant efforts in building new scene modelings are required.
>
> Thanks for the valuable suggestion. We have tested our proposed method on multiple settings, i.e., two tasks and two real-world datasets (HDD and nuScenes) for real-world scenarios. We sincerely agree that testing the proposed representations in a driving simulator is promising. We are building a similar data collection and labeling strategy in the CARLA simulator. We plan to test the proposed framework on end-to-end driving (e.g., the CARLA Challenge). We will leave it as future work.

---

> > ### Author Response · Authors · 2022-08-21
> > **Response to Reviewer 4hg8 (part 2 of 2)**
> >
> > **Q4** What is the clear definition of the semantic region?
> >
> > **A4** The birth of semantic region is motivated by road affordance (i.e., possible actions that a vehicle can take in an environment). For instance, a 4-way intersection affords left-turn, right-turn, and go-straight. We ask a question: how do we represent an afforded action? Existing strategies represent afforded actions via category (i.e., left-turn), trajectory, route, and destination/endpoint. As we know, there are infinite ways to take an afforded action. Our insight is that there are finite regions that a vehicle visits when taking action afforded the underlying road topology. Therefore, we define a vehicle will visit Region S (before passing the first crosswalk), Region A_i (on the first crosswalk), Region B_i (within an intersection), Region C_i (on the second crosswalk), and Region T_i (after passing the second crosswalk) in a 4-way intersection (shown in Figure 2 (a)).
> >
> > Given egocentric images recording a vehicle taking an afforded action, we apply COLMAP to obtain 3D scene structures, and camera poses. In addition, we apply a semantic segmentation algorithm to each egocentric image. With these processed data, we develop the semantic region labeling flow shown in Figure 3 to obtain the semantic region label of each egocentric image. The location of a camera pose determines the label. For instance, if the location of a camera pose is within an intersection, the corresponding semantic region label is B_i.

---

### Official Review · Reviewer_sBrF · 2022-08-01

**Originality:** Good
**Technical Quality:** Very Good
**Clarity Of Presentation:** Very Good
**Impact:** 3

**Recommendation:**

Weak Accept: I recommend accepting the paper, but will not argue for my recommendation if the majority of other reviewers have a different opinion.

**Summary:**

The authors propose a new method for predicting driver intent and risk objects from egocentric driving videos. Their approach relies on semantic region prediction based on road topologies. This approach improves performance on existing datasets.

**Issues:**

Long list of citations on line 20 contains duplicates.

In fig 1. The orange path is A1, B1, C1, T1, while the blue is A2, B2, B3, T2 and the green is A3, C2, C3, T3. So one is ABCT, the other two are ABBT and ACCT.

**Quality Of The Limitations Section:**

Limitations are addressed clearly

**Reviewer Expertise:**

4: The reviewer is confident but not absolutely certain that the evaluation is correct

**Robotics Focus:**

Highly relevant to robotics but no hardware experiments

**Strengths And Weaknesses:**

Strengths:

The problem is important and valuable.

The authors approach is interesting and makes sense.

The paper is easy to follow and understand.

The results show improvement in real world scenarios.

The authors use an automatic labelling scheme that may be of interest to the community.

Weaknesses:

The method only works with certain fixed road topologies, and does not handle unusual intersections and other edge cases that come up in the real world.

**Summary Of Recommendation:**

If the impact section had a 3.5: "possibly substantial impact in a paricular subfield" I would give this paper that. The results are good, the approach is interesting, if not revolutionary. Most importantly though, the content seems valuable to others working in this area.

---

> ### Author Response · Authors · 2022-08-21
> **Response to Reviewer sBrF**
>
> Thanks for the time and efforts that you have dedicated to reviewing our submission. We deeply appreciate your recognition that our work is valuable for people working in this area. In the following, we will address your concerns point-by-point in detail.
>
> **Q1** Duplicate citations
>
> **A1** Thanks for pointing out this issue. We have removed the duplicated citations in Line 20 of the main paper.
>
> **Q2** Wrong notations
>
> **A2** Thanks for your valuable comment. For Figure 1, the blue path should be S-A2-B2-C2-T2, and the green path should be S-A3-B3-C3-T3. We have updated the figure accordingly.
>
> **Q3** How can semantic regions handle unusual intersections and other edge cases that come up in the real world?
>
> **A3** Thanks for highlighting this important problem. The number of semantic regions of an intersection is determined by the number of branches of the intersection (also represents the number of afforded actions). Let us denote the number as n. A vehicle follows the order: S-Ai-Bi-Ci-Ti to complete an afforded action at an intersection. Therefore the total number of possible semantic regions is n*4+1. For a 4-way intersection, there are three different branches, and the total number of semantic regions is 13. For a 3-way intersection, the total number of semantic regions is 9 (two branches). Therefore, we can use this definition to define the corresponding semantic regions at unusual intersections such as 5-way or 6-way.
>
> For roundabout, we could define afforded actions of an ego vehicle as ‘stay in the lane’ and ‘exit the roundabout’, which can be considered as a series of Straight/Right-turn intersections in right-hand traffic countries or a series of Left-turn/Straight intersections in left-hand traffic countries.

---

> > ### Comment · Reviewer_sBrF · 2022-08-25
> > **Thanks**
> >
> > Makes sense, thanks!

---

### Official Review · Reviewer_wuu6 · 2022-08-02

**Originality:** Very Good
**Technical Quality:** Good
**Clarity Of Presentation:** Fair
**Impact:** 3

**Recommendation:**

Weak Accept: I recommend accepting the paper, but will not argue for my recommendation if the majority of other reviewers have a different opinion.

**Summary:**

This paper studies driver intent prediction and risk object identification from egocentric images/videos, which are crucial tasks for autonomous driving. This paper investigate a major problem: what's the ideal representation for these tasks? This paper proposes a novel scene representation, which is called semantic region. Such representation considers both higher-level semantic and geometric representations of roads. To learn this representation, this paper also introduces a novel semantic region prediction task. On the risk object identification task, this new representation achieves 6% improvements over existing best-performing methods. This paper studies an interesting problem for autonomous driving and proposes a novel scene representation. This novel representation can be potentially used for other tasks in autonomous driving and robotics.

**Issues:**

1. Since pre-training is very important as shown in multiple experiments, can authors provide more discussions of whether the comparisons to existing methods are fair?

2.  I encourage authors to add more details of semantic region since it's the major idea of this paper.

3. Can authors provide more ablation of pre-training on different datasets (nuScenes & Waymo Open Dataset)?

**Quality Of The Limitations Section:**

Additional details required

**Reviewer Expertise:**

4: The reviewer is confident but not absolutely certain that the evaluation is correct

**Robotics Focus:**

Highly relevant to robotics but no hardware experiments

**Strengths And Weaknesses:**

Overall, I am in favor of this paper because it proposes a novel representation for several important tasks in autonomous driving. Also, this paper shows significant improvements with this new representation on both driver intent prediction and risk object identification tasks. However, I do have several questions regarding this semantic region representation and some experiment details. I list strengths and weaknesses as follows.

Strengths:

1. This paper selects two important autonomous driving tasks to study. For these two tasks, this paper proposes a novel scene representation -- semantic region, which considers both higher-level semantics and geometrics of roads. Such representation is technically sound and will potentially benefit more tasks in autonomous driving/robotics.

2. This new representation is indeed effective. On the risk object identification task, this paper improves over best-performing methods by 6%. Also, on the driver intent prediction task, this new representation also exhibits very strong performance.

Weaknesses:

1. My major concern is the fairness of experiments. In data collection and annotation, this method uses a ResNet50 pre-trained on Mapillary Vistas dataset. I am worried that this actually introduces additional supervision/information for downstream applications studied in this paper.

2. I also find this paper is not easy to follow. This paper talks about semantic region as a new scene representation, however, it doesn't present what semantic region is in details. I would encourage authors to add more details of semantic region as it is the key idea of this paper.

3. In addition, the architecture shown in Figure 4 doesn't make sense to me. What are the roles of these two SR classifiers? Why are two classifiers are needed here?

4. In Section 3.2 (supplementary materials), pre-training on the in-domain data (Mapillary Vistas) is very important. Why is this method pre-trained on Mapillary Vistas instead of more popular autonomous driving datasets such as nuScenes/Waymo Open Dataset?

**Summary Of Recommendation:**

I think this paper presents interesting ideas and shows significant improvements with the new representation proposed in this paper, although there are certain weaknesses/problems with current draft. That being said, I am leaning towards weak acceptance if my concerns are addressed in the next revision.

---

> ### Author Response · Authors · 2022-08-20
> **Response to Reviewer wuu6 (part 1 of 2)**
>
> Thanks for your time and efforts in reviewing our submission. We appreciate your recognition that the proposed representation is novel for essential tasks in autonomous driving and additional valuable comments. In the following, we will address your concerns point-by-point in detail.
>
> **Q1**: Fairnesses of Experiments
>
> **A1**: Thanks for pointing out the critical issue. The primary goal of this work is to answer the question, is semantic region an effective road scene representation for downstream traffic scene tasks? Toward the goal, we construct a dataset with semantic region annotations. We use COLMAP (SLAM) and a semantic segmentation algorithm pre-trained on the Mapillary dataset to enable the labeling of semantic regions. Note that new information is indeed introduced. Then, the labels are used to train the proposed semantic region prediction to obtain road scene representations.
>
> For intent prediction, all the models share the same feature extractor (ResNet50 pre-trained on the Mappilary dataset) and are trained on the same dataset to ensure fairness. We show that methods with auxiliary loss (LSTM-EL), advanced architecture (OadTR), and trajectory supervision (TRN-Tra) cannot achieve favorable results. SRI-INT demonstrates significant improvements compared with baselines. Table 2 serves as empirical evidence to the question we raised.
>
> For risk object identification, we use the same feature extractor (i.e., InceptionResnet-V2 pre-trained on ImageNet) as the baseline. To further justify whether the proposed task relies on the “good features,” we further train representations using features pre-trained on ImageNet. The results are shown in the 3rd row in both tables below. We found that semantic region prediction indeed learns informative representations for risk object identification.
>
> Table for Crossing Vehicle
>
> |           Model          | Acc 0.5 | Acc 0.75 | mAcc |
> |:------------------------:|:-------:|:--------:|:----:|
> |     [27] (paper)           |   49.2  |   48.6   | 43.0 |
> | [27] (our implementation) |   49.2  |   48.2   | 42.7 |
> |     SRP-ROI_ImageNet     |   50.8  |   49.8   | 43.9 |
> |     SRP-ROI_Mapillary    |   51.8  |   51.1   | 45.1 |
>
>
> Table for Crossing Pedestrian
> |           Model           | Acc 0.5 | Acc 0.75 | mAcc |
> |:-------------------------:|:-------:|:--------:|:----:|
> |        [27] (paper)       |   35.7  |   32.1   | 27.0 |
> | [27] (our implementation) |   33.3  |   29.8   | 26.2 |
> |      SRP-ROI_ImageNet     |   38.1  |   34.5   | 30.1 |
> |     SRP-ROI_Mapillary     |   42.9  |   39.3   | 33.3 |
>
> **Q2** Can the authors add more details of semantic region?
>
> **A2** The birth of semantic region is motivated by road affordance (i.e., possible actions that a vehicle can take in an environment). For instance, a 4-way intersection affords left-turn, right-turn, and go-straight. We ask a question: how do we represent an afforded action? Existing strategies represent afforded actions via category (i.e., left-turn), trajectory, route, and destination/endpoint. As we know, there are infinite ways to take an afforded action. Our insight is that there are finite regions that a vehicle visits when taking action afforded the underlying road topology. Therefore, we define a vehicle will visit Region S (before passing the first crosswalk), Region A_i (on the first crosswalk), Region B_i (within an intersection), Region C_i (on the second crosswalk), and Region T_i (after passing the second crosswalk) in a 4-way intersection (shown in Figure 2 (a)).
>
> **Q3** What are the roles of these two SR classifiers?
>
> In this work, we aim to demonstrate that the proposed representation can be applied to diverse road topologies (i.e., 4-way, 3-way, and straight road). Different topologies need a different set of semantic regions, as shown in Figure 2. To have a unified framework, we first use a topology classifier to determine whether the ego-vehicle is at intersections (4-way, 3-way) or non-intersections (straight road). We design two SR classifiers to predict the corresponding set of semantic regions.
>
> Our design is similar to CILRS [1], a command-conditional imitation learning framework. In our case, we select the corresponding set of semantic regions (as shown in Figure 1) based on the prediction of topology type.
>
> [1] Codevilla et al., End-to-end Driving via Conditional Imitation Learning, ICRA 2018

---

> > ### Author Response · Authors · 2022-08-20
> > **Response to Reviewer wuu6 (part 2 of 2)**
> >
> > **Q4** The impact of pre-training features
> >
> > **A4** When we train the semantic region predictor, we fix the backbone (i.e., ResNet50 pre-trained on Mapillary). We chose features pre-trained on Mapillary because we want to capture road scene structure. An ImageNet pre-trained backbone does not see road scenes in the pretraining stage. Therefore, we observe inferior intent prediction results, as shown in the supplementary material.
> >
> > The Mapillary Vista dataset is a challenging road scene segmentation benchmark, and it provides a fine-grained pixel-level annotation to facilitate road scene understanding. The nuScenes dataset and Waymo Open Dataset are popular autonomous driving datasets, but they focus on detection, tracking, 3D semantic segmentation, and future trajectory prediction. The recent release (v1.4.0) of the Waymo Open Dataset (Perception) offers annotations for video panoptic segmentation. Due to the limited time and the size of the dataset, we do not train a segmentation model using the dataset. Note that it is of great interest to train on the Waymo video panoptic segmentation dataset in our future work. In the next post, we will present additional results about backbone pre-training.

---

> > > ### Author Response · Authors · 2022-08-26
> > > **More Ablation of Pre-training on Different Datasets**
> > >
> > > As mentioned in the previous section, nuScenes and Waymo Open Dataset focus on detection, tracking, 3D semantic segmentation, and future trajectory prediction. The nuScenes dataset provides BEV map segmentation labels, but such information is not pixel-accurate for semantic segmentation or panoptic segmentation in perspective-view images. The very recent v 1.4.0 release of the Waymo Open dataset (Perception) includes 2D video panoptic segmentation labels that are potentially useful for our tasks.
> > >
> > > The Mapillary backbone is pre-trained on the panoptic segmentation task. We provide additional ablations of the backbone of SRI_INT in the following tables. The nuScenes backbone is trained on instance segmentation using data released in nuImage, an extension of nuScenes that contains additional images and 2D annotations. Note that they have semantic labels for drivable surface. The COCO Panoptic backbone is trained on COCO Panoptic Segmentation. The model performs favorably on different settings, while COCO Panoptic is not a traffic scene dataset. The table shows that the in-domain nuScenes backbone cannot perform well in many metrics. We hypothesize that the two tasks, i.e., intention prediction and risk object identification, require “Stuff” information (e.g., road, lane marking, and crosswalk). On the other hand, the nuScenes backbone learns to detect objects, which could explain the superior performance on HDD_interactive_cases because these cases involve interaction with other traffic participants.
> > >
> > > Note that we do not have models pre-trained on the Waymo Open dataset because there are no public pre-trained weights due to the dataset policy. In addition, we have limited time to train a model on such a large-scale dataset and finetune our task. We leave a comparison with a backbone trained on the Waymo Open dataset as future work.
> > >
> > > Table for HDD
> > >
> > > |    Feature    |  Macro Precision | Micro Precision |  mAP |
> > > |---------------|:---------------:|:---------------:|:----:|
> > > |    ImageNet   |       51.3      |       74.6      | 53.9 |
> > > |    nuScenes   |       25.0      |       57.8      | 45.5 |
> > > | COCO Panoptic |       52.8      |       61.8      | 51.7 |
> > > |   Mapillary   |       55.3      |       73.8      | 57.9 |
> > >
> > > Table for HDD Interactive
> > >
> > > |   Feature  | Marco Precision | Micro Precision |  mAP |
> > > |---------------|:---------------:|:---------------:|:----:|
> > > |    ImageNet   |       45.3      |       59.3      | 60.1 |
> > > |    nuScenes   |       67.4      |       69.8      | 69.1 |
> > > | COCO Panoptic |       46.1      |       61.1      | 63.3 |
> > > |   Mapillary   |       67.0      |       70.3      | 69.5 |
> > >
> > > Table for nuScenes
> > >
> > > |   Feature  | Macro Precision | Micro Precision |  mAP |
> > > |---------------|:---------------:|:---------------:|:----:|
> > > |    ImageNet   |       36.0      |       59.7      | 58.1 |
> > > |    nuScenes   |       45.1      |      37.6      | 59.5 |
> > > | COCO Panoptic |       37.6      |       63.8      | 61.1 |
> > > |   Mapillary   |       41.1     |       68.3      | 66.7 |

---

### Official Review · Reviewer_1ZBN · 2022-08-04

**Originality:** Fair
**Technical Quality:** Good
**Clarity Of Presentation:** Good
**Impact:** 2

**Recommendation:**

Weak Reject: I recommend rejecting the paper, but will not argue for my recommendation if the majority of other reviewers have a different opinion.

**Summary:**

The paper proposes a computer vision pipeline for computing semantic regions in a scene and predicting where the car will go in terms of those regions, in the context of autonomous vehicles. The method is evaluated using publicly available datasets and compared to other methods for the same or similar problems.


**Issues:**


There are a number of clarity issues in the paper.

The sentence "While promising results are demonstrated, the learned representations are effective to the trained task" is not clear, do you mean to say ineffective? Or something else?

The term "semantic region" is defined very narrowly and hence, it would be better to use a different, more specific term that relates to a region that the car will be in the future.

Also, is there a particular reason why map information is not used as an additional input to the model?

Some references are messed up, for example [3]

**Quality Of The Limitations Section:**

Limitations are addressed clearly

**Reviewer Expertise:**

3: The reviewer is fairly confident that the evaluation is correct

**Robotics Focus:**

Relevant but unlikely to deploy to hardware in near future

**Strengths And Weaknesses:**


As this is a computer vision paper, it is difficult for me to judge particular strengths.

The main weakness with respect to the venue is that this is a computer vision paper being submitted to a robotics conference.

In addition, the authors need to motivate better why predicting the future in terms of discrete semantic regions is better than just predicting a continuous trajectory, which is what many other works do.



**Summary Of Recommendation:**


My main justification is that this paper belong in a computer vision venue or at a venue that solely focuses on autonomous driving.

---

> ### Comment · Reviewer_sBrF · 2022-08-18
> **vision-for-self-driving is robotics**
>
> I'm not sure if this is my place as another reviewer, but I would gently argue that vision-for-self-driving is a robotics problem, and is an appropriate fit for this conference.  Other vision-for-self-driving papers have been well received here in the past.

---

> ### Author Response · Authors · 2022-08-20
> **Response to Reviewer 1ZBN  (part 1 of 2)**
>
> Thanks for your time and efforts in reviewing our submission and the valuable comments. In the following, we will address your concerns point-by-point in detail.
>
> **Q1**: How is this paper related to robotics?
>
> **A1**: Vision-based intelligent driving systems such as autonomous driving and advanced driver assist systems are robotics problems. Recently, we have seen vision-based solutions deployed by companies such as Wayve, Tesla, and Mobileye. Moreover, vision-based algorithms for intelligent driving systems have been well-perceived in the past CoRL conferences (e.g., [1, 2, 3, 4] as shown below).
>
> In this work, we propose a novel representation called a semantic region. Then, we propose to learn road scene representation via semantic region prediction. The learned representation aims to facilitate downstream tasks (identify intents and potential hazards in this work) critical for intelligent driving systems. Therefore, we want to gently argue that the work is not a computer vision paper but a paper aiming to advance vision-based intelligent driving systems.
>
> [1] Sauer et al., Conditional Affordance Learning for Driving in Urban Environments, CoRL 2018
>
> [2] Chen et al., Learning by Cheating, CoRL 2019
>
> [3] Xiao et al., Action-based Representation Learning for Autonomous Driving, CoRL 2020
>
> [4] Scheel et al., Urban Driver: Learning to Drive from Real-world Demonstrations Using Policy Gradients, CoRL 2021
>
>
> **Q2**: why map information is not used?
>
> **A2**: Thank you for pointing out the important modality. We would like to emphasize that a map-based solution is not the scope of this work. As mentioned in Q1, we aim to develop a vision-based intelligent driving system with egocentric cameras as inputs. Given limited information, advanced traffic scene modeling becomes indispensable. It is worth noting that map information is vital for scene understanding and planning. Therefore, we motivate by well-known mapping modules, i.e., vision-based SLAM algorithm (COLMAP) and semantic segmentation, to construct ground truth labeling of semantic regions. We propose to learn representations via semantic regions prediction and test the learned representations on multiple real-world traffic scene datasets (HDD and nuScenes) and tasks (intent prediction and risk object identification).
>
> **Q3**: Why not predict trajectory?
>
> **A3**: We sincerely agree with the reviewer that trajectory should be studied. Trajectory is a popular representation that the community has studied for a long time. Our insight is that semantic regions capture a holistic view of scenes, encode feasible destinations, and specify afforded actions given a topology. On the other hand, pure trajectories capture a short-term movement of an agent. Recent trajectory forecasting algorithms (e.g., [5, 6]) leverage the concept of target/endpoint to improve forecasting performance. These works are relevant to the intent of semantic regions. However, there are two differences are observed:
> The prediction algorithms are performed given observed states in BEV [5, 6]. In contrast, inputs to the proposed system are egocentric images. Therefore, we proposed a labeling strategy to generate the corresponding semantic regions and a framework to learn road scene representations for downstream tasks.
> The goal of [5,6] is to improve forecasting performance instead of learning road scene representations for downstream tasks.
> We will incorporate the discussion into the manuscript to emphasize differences.
>
> [5] Zhao et al., “TNT: Target-driveN Trajectory Prediction,” CoRL 2020
>
> [6] Mangalam et al., “It Is Not the Journey but the Destination: Endpoint Conditioned Trajectory Prediction,” ECCV 2020
>
> Given the relevance to trajectory prediction, it motivates us to construct a baseline predicting future trajectories (obtained from projecting camera poses calculated by COLMAP onto egocentric images). Our experiments show that predicting semantic regions demonstrates significant improvements (10+% in mAPs) over predicting future trajectories on multiple settings, as shown in Table 2.

---

> > ### Author Response · Authors · 2022-08-20
> > **Response to Reviewer 1ZBN (part 2 of 2)**
> >
> > **Q4**: Is "semantic region" the best term?
> >
> > **A4**: We currently cannot identify a better term than “semantic region.” “Semantic” means that our representation aims to capture semantic structures of a traffic scene, and “Region” is the semantic structure. We then use a sequence of semantic regions to represent an afforded action, as shown in Figure 2. Indeed, these regions are associated with regions where the vehicle will be in the future. It is worth noting that the concept of "a region will be" is related to intention and affordance, which is not the goal of the proposed representation. We would be deeply grateful if we could have a conversation with Reviewer 1ZBN in the discussion phase to shape the naming jointly.
> >
> > **Q5**: "While promising results are demonstrated, the learned representations are effective to the trained task" is not clear. Do you mean to say ineffective? Or something else?
> >
> > **A5**: Thanks for pointing out the unclear statement. Yes, we meant to say the existing learned representations are ineffective.
> >
> > **Q6** Incorrect reference format
> >
> > **A6** Thanks for pointing out the issue. We have updated the reference accordingly.

---

### Meta-Review · Area_Chair_AHrw · 2022-08-14

**Recommendation:** Accept (Poster)
**Confidence:** 4

**Metareview:**

Reviewers like the novel formulation of the method, but are concerned about the fairness of the experiments, the generalization to unseen topologies, and the presentation.  One reviewer also argued that the topic is not closely relevant to robotics, which is required by CoRL.  Please consider addressing these points if possible.

The authors did a good job during rebuttal, after which three of the four reviewers were in favor of acceptance.  The main concern of the negative reviewer was the submission's alignment with CoRL.  The AC agrees that this is a reasonable concern, but also well addressed by the authors and other reviewers.  In this case, the AC is recommending acceptance.  The authors should revise the submission accordingly based on the reviews.

**Best Paper Nomination:**

No

---

> ### Author Response · Authors · 2022-08-20
> **Response to Area Chair AHrw**
>
> We thank you and all the reviewers for reading our submission and providing constructive feedback. We do our best to provide point-by-point responses to the reviewers’ comments and concerns.